# Randomized controlled trial parallel-group on optimizing community pharmacist's care for the elderly: The influence of WhatsApp-Email delivered clinical case scenarios

Osama Mohamed Ibrahim[1,2], Nadia Al Mazrouei[3], A. A. Elnour[4,5], Rana Ibrahim[3], Derar H. Abdel-Qader[6], Rowida Mohamed El Amin Ibrahim Hamid[7,8], Vineetha Menon[9], Ali Awadallah Saeed[10]*, Sami Fatehi Abdalla[11], Fahad T. Alsulami[12], Yousef Saeed Alqarni[13], Abuelnor Mohammed[14,15]

1 Department of Clinical Pharmacy, School of Pharmacy, New Giza University, 6th of October City, Egypt, 2 Department of Pharmacy Practice and Pharmacotherapeutics, College of Pharmacy, University of Sharjah, Sharjah, United Arab Emirates, 3 Department of Pharmacy Practice and Pharmacotherapeutics, Faculty of Pharmacy, University of Sharjah, Sharjah, United Arab Emirates, 4 Program of Clinical Pharmacy, College of Pharmacy, Al Ain University, Abu Dhabi Campus, Abu Dhabi, United Arab Emirates, 5 AAU Health and Biomedical Research Center, Al Ain University, Abu Dhabi, United Arab Emirates, 6 Faculty of Pharmacy and Medical Sciences, University of Petra, Amman, Jordan, 7 Program of Pharmacotherapy, National University, Khartoum, Sudan, 8 Department of Pharmacology, Al-Fajr University, Mianwali, Sudan, 9 Department of Pharmacy Practice, College of Pharmacy, Gulf Medical University, Ajman, UAE, 10 Department of Pharmacology, Faculty of Clinical and Industrial Pharmacy, Mycetoma Research Center, National University, Khartoum, Sudan, 11 Clinical Department, College of Medicine, University of Almaarefa, Diriyah, Riyadh, Saudi Arabia, 12 Clinical Pharmacy Department, College of Pharmacy, Taif University, Taif, Saudi Arabia, 13 Department of Pharmacy Practice, College of Pharmacy, Imam Abdulrahman Bin Faisal University, Dammam, Saudi Arabia, 14 Department of Basic Medical Sciences, College of Medicine, Dar Al Uloom University, Riyadh, KSA, 15 Department of Histology and Embryology, School of Basic Medical Sciences, Tongji Medical College, Huazhong University of Science and Technology, Wuhan, People's Republic of China

* alimhsd@gmail.com

## Abstract

### Background

Drug-related problems (DRPs) significantly threaten the safety of the elderly. In order to improve pharmacists' ability to minimize these events, novel educational interventions that consider the current challenges in clinical practice are crucial.

### Objectives

The primary objective is to assess the impact of two unique educational interventions on pharmacists' ability to identify DRPs.

### Method

A total of 127 community pharmacies in the United Arab Emirates (UAE) were recruited and randomly allocated to one of three arms using a 1:1:1 allocation ratio. While a series of clinical case scenarios (N = 24) related to elderly care were established and validated and sent to pharmacists in Active Group A over a 3-month period, lengthy research articles focused

**Data Availability Statement:** All relevant data are within the manuscript and its Supporting Information files.

**Funding:** The author(s) received no specific funding for this work.

**Competing interests:** The authors have declared that no competing interests exist.

on medication safety in elderly were emailed to pharmacists in Active Group B. The control group pharmacist received no intervention. Then, pharmacists self-reported the number, categories, and severity of DRPs and pharmacist recommendations.

## Results

The incidence of DRPs identified by pharmacists was 10.8% in Active Group A, 2.0% in the Control Group (p = 0.011), and 3.8% in Active Group B (p = 0.014). A significant difference was observed in the proportion of DRP types between Active Group A and the Control Group. The most common DRPs in Active Group A were avoidable medication (14.7%) and untreated disease (9.2%). Pharmacists in Active Group A (37.2%) and Active Group B (32.3%) most commonly intervened by recommending the cessation of medication, while the most common intervention in the Control Group was recommending a decrease in dose (29.8%). The mean cost reduction per patient was highest in Active Group A (31.3 ±11.8 $), followed by Active Group B (20.8 ±8.6 $) and the Control Group (19.6 ±9.5 $). The mean time needed to resolve a DRP was shortest in Active Group A (7.3 ±3.5 minutes), followed by Active Group B (9.8 ±4.2 minutes) and the Control Group (9.8 ±5.7 minutes).

## Conclusion

Using WhatsApp to deliver clinical scenarios was effective in improving pharmacists' ability to identify and address DRPs in elderly patients, resulting in faster resolution and higher cost savings.

## Introduction

Drug-related problems (DRPs) pose a significant threat to older individuals' health and quality of life [1, 2]. With high rates of chronic conditions and multiple medication use, older individuals are more prone to adverse drug events (ADEs) and hospitalization, leading to decreased quality of life and higher healthcare costs [3]. Their vulnerability to DRPs is due to aging-related changes in pharmacokinetics, pharmacodynamics, and kidney/liver function, as well as cognitive decline and mobility issues [4, 5]. There was a large difference globally in the prevalence of DRPs among older people, ranging from 14.1% (USA) to 95.9% (the Netherlands) [6].

Community pharmacists have a vital role in safeguarding the health of elderly and mitigating the risk of DRPs [7–9]. As medication experts, they are capable of conducting medication reviews and providing advice on safe and appropriate medication use, which can positively impact the health outcomes of elderly [10]. Furthermore, community pharmacists can educate patients on safe medication practices, including proper storage, administration, and an understanding of potential side effects and adverse reactions [10, 11]. However, despite the importance of their role, community pharmacists may encounter challenges in executing their duties effectively. These challenges may include limited time, inadequate resources, or insufficient training. These constraints can obstruct pharmacists from providing comprehensive medication reviews and education to patients, leading to an increased risk of DRPs and adverse health outcomes. To enhance their knowledge, skills, and competencies in managing DRPs among elderly, education is critical for pharmacists. However, traditional educational methods are unrealistic because of the high costs and time demands, and they tend to struggle with keeping learners engaged, especially over an extended period [12, 13].

Therefore, the use of technology-based tools, such as WhatsApp, as a vehicle for educational interventions in the healthcare field has been increasing in recent years. Despite the distraction of health professionals and the eye constrain upon prolonged use [14], The benefits of using WhatsApp include improved information distribution, communication, and accessibility [12, 15]. WhatsApp has been found to be a useful tool for communicating and distributing information among healthcare professionals [9, 15–18].

## Rationale

Previous research has shown that the use of clinical case scenarios as an assessment tool is effective. However, its impact on enhancing the knowledge and abilities of healthcare professionals, particularly pharmacists, has yet to be explored.

## Objectives

The primary objective is to assess the impact of two unique educational interventions on pharmacists' ability to identify DRPs. The current study aims to investigate the effectiveness of two interventions on the safety of older patients in community pharmacies. The first intervention is the use of clinical case scenarios related to older care sent to community pharmacists via WhatsApp. The second intervention is the use of lengthy articles related to older patient care sent via email. The study will examine the impact of these interventions on pharmacists' ability to identify drug-related problems, perform medication reviews, and perform pharmaceutical interventions.

## Methods

### Study design and sample size calculation

This 9-month, parallel-group, randomized trial was conducted in the United Arab Emirates between March 2022 and January 2023, and involved 127 community pharmacies, which were randomly assigned to one of three groups using a 1:1:1 allocation ratio. Active Group A received 24 clinical case scenarios related to elderly care, delivered via WhatsApp over a 12-week period. Active Group B received 24 research articles focused on medication safety for elderly, sent via email over the same 12-week period. The control group consisted of pharmacies that did not receive any intervention. The frequency, types, and severity of drug-related problems and pharmaceutical interventions in elderly' medication therapy plans were self-reported using a standardized data collection sheet over a 6-month period. The study was structured in accordance with the Consolidated Standards of Reporting Trials (CONSORT 2010), as outlined in **S1 File**.

The sample size was calculated using the equation: $n = (Z^2 \times p \times (1 - p)) / E^2$ [19], where Z represents the Z-score corresponding to a 95% confidence level (1.96), p is the baseline incidence rate (0.0046), and E is the desired margin of error (0.00092). The calculation assumed a 20% improvement in pharmaceutical interventions for elderly medication therapy plans, including the detection of drug-related problems (DRPs). With these values, the sample size was estimated to be 141 per group, rounded up from 140.95. An additional 10% was added to account for an expected attrition rate of 10%, resulting in a final sample size of 156 pharmacists per group and a total sample size of 468 for the study.

The study received ethical approval from the University of Sharjah Ethics Committee.

### Study procedure

The principal investigator (OMI) created a digital list of community pharmacies, including their names, phone numbers, emails, and locations. The pharmacies were called and evaluated

for eligibility based on the following criteria: operating hours (open at least 5 days a week with at least two shifts), employment of at least three pharmacists, and providing services to at least 10 elderly customers per day. Pharmacies located more than 15 miles from a clinic or hospital, and those currently participating in another study, were excluded. Next, pharmacists were recruited based on specific inclusion criteria, including at least two years of experience in community pharmacy, regular working days, using WhatsApp, and proficiency in English or Arabic. Those who had participated in training workshops or programs focused on elderly care or medication safety were excluded. The managers of eligible pharmacies were then contacted and asked to sign a consent form, and the eligible pharmacists were approached to participate in the study. A pharmacy's inclusion in the study was contingent on the agreement of both its manager and staff.

The randomization process was performed at the pharmacy level using a blind procedure to prevent knowledge of the assignment from being disclosed to the participants, data operators, and the research team. This was accomplished using the random number generator in SPSS version 26. For active Group A, an expert panel of two clinical and one community pharmacists created 24 clinical case scenarios of elderly patients with Drug-Related Problems (DRPs). The cases, with a maximum word count of 150, were based on the panel members' experiences and previous literature [20]. The scenarios consisted of 36 prescriptions, 61 medications, and 49 DRPs. The patient's demographics (age, sex), past medical history, diagnosis, comorbidities, and laboratory results (if necessary) were briefly described in each case. The DRPs were then identified and a pharmacist intervention was proposed. The cases were reviewed and validated by three independent assessors: a general practitioner, a community pharmacist, and a geriatrician.

Polypharmacy is a common issue among elderly and is a major risk factor for DRPs [21]. To address this, twelve case scenarios were developed for patients who take multiple medications, which can increase the likelihood of adverse drug reactions and interactions. For instance, "Mr. Khalid, a 75-year-old male, sought Naproxen for his back pain but was already taking Lisinopril, metformin, simvastatin, omeprazole, and aspirin". Elderly may struggle with complex dosing regimens, leading to incorrect dosage due to issues such as decreased vision [22]. Hence, three cases were designed to address dosing-related issues in these patients. Over-prescribing is also a common cause of DRPs in elderly, thus, "deprescribing" or discontinuing unnecessary medications was proposed in four cases [23, 24]. For example, "Mrs. Saad, an 85-year-old female patient, had low blood pressure (80/50 mmHg) and elevated creatinine (2.5 mg/dL) during her check-up while taking lisinopril 20 mg, simvastatin 40 mg, aspirin 81 mg, alprazolam 1 mg, and acetaminophen 650 mg".

A community group was created on WhatsApp, where the main investigator posted two clinical case scenarios per week for three months. Only participants in Active Group A, who could only see texts from the admin, were added to the group. For Active Group B, the research team conducted a search of the Cochrane Library for articles published after 2015 that documented DRPs, potentially inappropriate medications, and adverse drug reactions in elderly. After evaluating potential articles through three rounds, 24 articles were selected and sent to participants in PDF format, two articles per week for duration of three months. The data collection sheet also recorded the total number of elderly and their characteristics, such as the presence of major polypharmacy and the number of prescriptions and medication orders. DRP classification was based on the AbuRuz index [25]. The severity of DRPs was evaluated by a multidisciplinary expert panel, which met twice a month during the data reporting period and reviewed and rated each DRP reported by the participants (Fig 1).

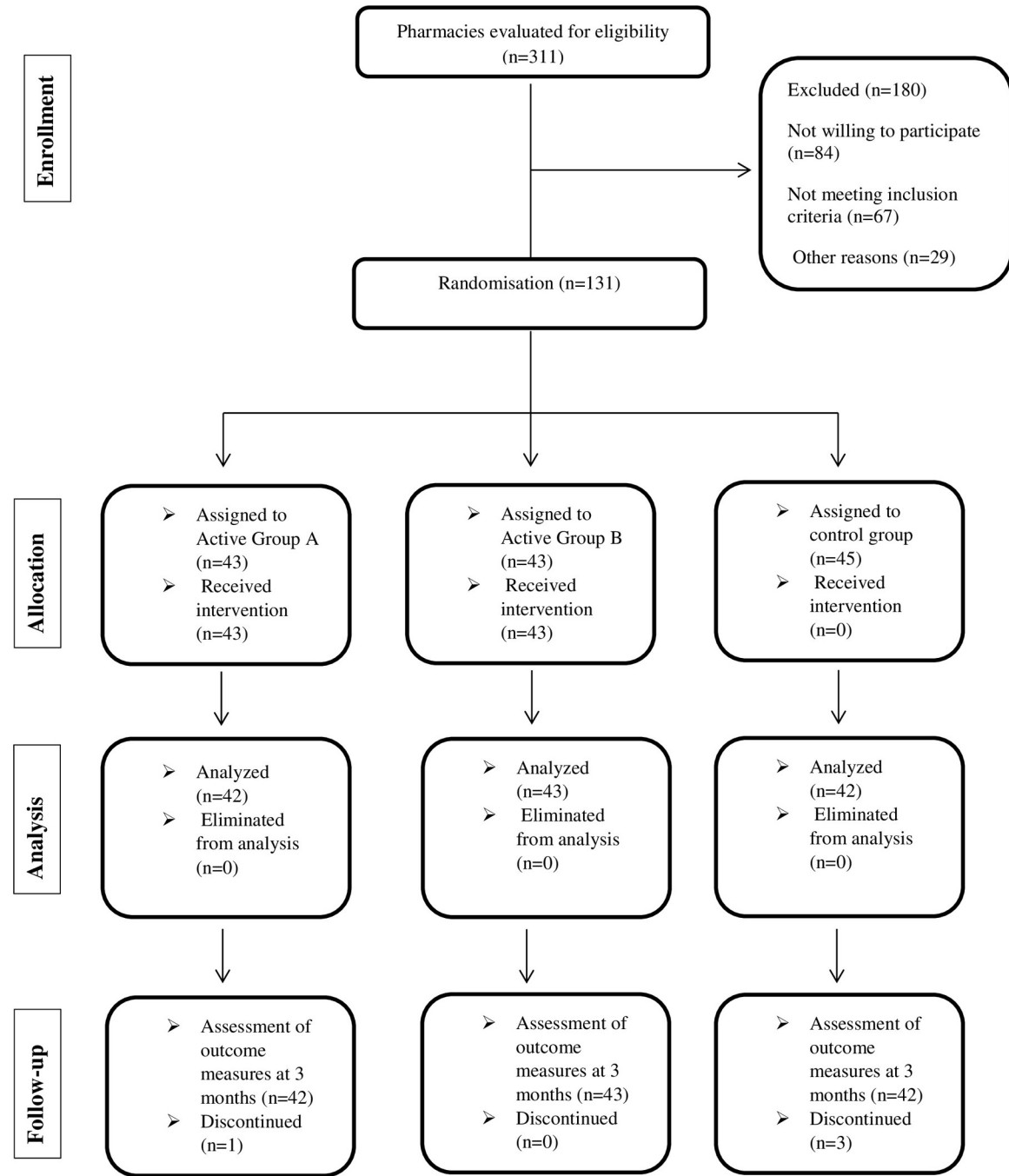

**Fig 1. The flow diagram of the study (CONSORT, 2010).**

## Outcome measure

The primary outcome measure was the improvement in the ability to identify DRPs. The differences in means between the interventions groups were taken as the outcome measure (the incidence of DRPs identified by pharmacists). The primary and secondary outcomes of the study were assessed using an electronic data collection sheet, delivered to pharmacists in all study groups and filled out over six months. The primary outcomes included the incidence,

categories, and severity of DRPs, as well as the frequency and types of pharmacist interventions at both the physician and patient levels. The secondary outcomes included cost reduction per patient and time required to resolve a DRP.

## Statistical analysis

The data analysis was performed using SPSS version 26.0. The Shapiro-Wilk test was used to assess the normality of the data, which showed normal distribution (p = 0.086). One-way ANOVA was applied to test the differences in means among the study groups, followed by post-hoc analysis for two-group comparisons. The proportion of categorical variables such as DRP incidence, classification, and severity were compared across study groups using Chi-square test. Results were considered statistically significant if the p-value was below 0.05. Finally, error-bars were used to compare the mean cost reduction per patient and the mean time to resolve a DRP across the study groups. Continuous variables were presented as mean and standard deviation, while categorical variables were represented as numbers and percentages.

## Results

A total of 468 community pharmacists working in 131 pharmacies agreed to participate in the study and were assigned to three groups; the control group, Active Group A, and Active Group B. However, 30 pharmacists and four pharmacies withdrew from the study for unknown reasons. The majority of the pharmacies in the study were females and had access to a permanent internet connection. A notable difference between the groups is that 39.1% of the pharmacies in the control group provided telepharmacy services, while 41.9% and 45.2% of the pharmacies in Active Groups A and B provided telepharmacy services, respectively. On average, pharmacists in Active Group A used WhatsApp slightly more frequently per day than in other groups. Overall, there were no significant differences in the characteristics of the pharmacies or pharmacists among the study groups, as shown in **Table 1**.

The total number of elderly handled by pharmacists in the control group, Active Group A, and Active Group B was 3581, 3348, and 3255, respectively (**Table 2**). There were no significant differences in the numbers of elderly with major polypharmacy, elderly living alone, and elderly requiring help in daily activity among the study groups. However, pharmacists in Active Group A identified a significantly higher number of elderlies with DRPs (n = 689) compared to the control group (n = 214) and Active Group B (n = 318). The incidence of DRPs in Active Group A was 10.8% versus 2.0% in the control group (p = 0.011) and 3.8% in Active Group B (p = 0.014). There was no significant difference in the incidence of DRPs between Active Group B and the control group.

In terms of DRP categories, there were significant differences in the categories of DRPs between the Active Group A and the control group. Specifically, the most common type detected by pharmacists in Active Group A was avoidable medicine (14.7%) followed by untreated disease (9.2%). In contrast, the need for additional treatment (10.5%) and contraindication (9.6%) were the most common types of DRPs identified by pharmacists in the control group. The proportions of DRP categories between Active Group B and the control group (p = 0.072) and between Active Groups B and A (p = 0.081) were similar.

With regard to the severity of DRPs and the evidence quality, there were no statistically significant differences among the study groups. The majority of DRPs were mild (66.4% in the control group, 56.2% in Active Group A and 57.0% in Active Group B), followed by significant (25.0% in the control group, 33.7% in Active Group A and 34.4% in Active Group B) and serious (7.5% in the control group, 8.1% in Active Group A and 7.0% in Active Group B). The least

**Table 1. Baseline characteristics of the study sample[β][μ].**

| Pharmacy | Control group | Intervention group A | Intervention group B |
|---|---|---|---|
| Number of pharmacies, n | 42 | 42 | 43 |
| Pharmacy size in square feet | 480.6 (±32.7) | 477.6 (±33.5) | 481.1 (±34.9) |
| Permanent internet connection, yes, n (%) | 38 (90.5%) | 39 (90.7%) | 40 (95.2%) |
| Providing remote pharmacy services | 16 (38.9%) | 18 (41.9%) | 19 (45.2%) |
| The average number of onsite patients per day | 47.3 (±12.5) | 45.2 (±9.1) | 48.8 (±13.3) |
| The average number of remote patients per day | 13.2 (±8.6) | 11.9 (±7.4) | 13.5 (±9.3) |
| The average number of onsite elderly patients per day | 11.8 (±3.5) | 12.2 (±4.2) | 11.7 (±4.6) |
| The average number of remote elderly patients per day | 4.7 (±2.4) | 5.9 (±3.4) | 5.5 (±2.8) |
| The average number of prescriptions filled for elderly patients per day | 8.9 (±6.5) | 8.1 (±5.4) | 7.5 (±4.8) |
| The average number of prescribed medications dispensed for elderly patients per day | 38.7 (±9.2) | 37.5 (±9.6) | 35.4 (±11.0) |
| The average number of non-prescribed medications dispensed for elderly patients per day | 31.2 (±7.5) | 28.5 (±8.2) | 32.4 (±7.9) |
| **Pharmacist** | | | |
| Number of pharmacists, n | 144 | 148 | 146 |
| Sex, female, n (%) | 112 (77.8%) | 117 (79.1%) | 115 (78.8%) |
| Age, (years) | 28.7 (±5.2) | 27.4 (±4.3) | 26.9 (±3.9) |
| Experience, (years) | 5.3 (±2.8) | 4.9 (±1.9) | 5.2 (±2.3) |
| Degree, n (%) | | | |
| BSc | 123 (85.4%) | 130 (87.8%) | 127 (87.0%) |
| MSc | 16 (11.1%) | 12 (8.1%) | 15 (10.3%) |
| PharmD | 5 (3.5%) | 6 (4.1%) | 4 (2.7%) |
| Attended a PDP specialised in elderly care | 21 (14.6%) | 18 (12.2%) | 20 (13.7%) |
| Attended a PDP specialised in medication safety | 28 (19.4%) | 26 (17.6%) | 31 (21.2%) |
| *Average daily WhatsApp usage, hours | 4.5 (±2.7) | 5.3 (±3.6) | 4.6 (±2.9) |

Variables are presented as mean and standard deviation (SD) unless stated otherwise. PDP: professional development programme.

* Data were accessed from the smartphone of each pharmacist. Differences in means across the study arms were calculated using the one-way ANOVA test and differences in proportions were measured using the Chi-square test.

[β]Pharmacy and pharmacists' characteristics were similar across the study arms (p>0.05).

μ: The average numbers of patients, medications, and prescriptions of each pharmacy were measured a month before initiating the study.

common severity of DRPs was life-threatening (1.2% in the control group, 1.9% in Active Group A and 1.5% in active group B). In terms of evidence quality, the majority of DRPs were moderate (45.8% in the control group, 46.5% in Active Group A and 47.0% in Active Group B).

While pharmacists conducted 394 interventions in the control group, of which 43.4% (n = 171) were at the physician level, the total number of pharmacist interventions performed by pharmacists in Active Group A and group B was 1629 and 536, respectively, of which 47.2% (n = 769) and 37.5% (n = 201) were at the physician level, respectively (**Table 3**). There were no statistically significant differences in the proportions of categories of pharmacist interventions at the physician (p = 0.086) and patient levels (p = 0.094) among the study groups. There were six intervention categories at the physician level and nine intervention categories at the patient level. At the physician level, recommending cessation of medication was the most common intervention made by pharmacists in Active Group A (37.2%) and Active Group B (32.3%), whereas, recommending decreasing a dose (29.8%) was the most common intervention in the control group. At the patient level, patient education was the most common intervention made by pharmacists in Active Group A (29.2%), group B (35.5%), and the control group (28.3%). Followed by referral to a prescriber in Active Group A (13.3%) and the control group (13.9%), and recommending a monitoring plan in Active Group B (12.2%).

**Table 2. Incidence, classification, and severity of drug-related problems (DRPs) reported by the study arms.**

| Variable | Control group (C) | Intervention group A | Intervention group B | P value[a] (C vs A) | P value[b] (C vs B) | P value[c] (A vs B) |
|---|---|---|---|---|---|---|
| Number of older patients | 3581 | 3348 | 3255 | 0.512 | 0.433 | 0.619 |
| Number of older patients with major polypharmacy | 1482 | 1559 | 1365 | 0.163 | 0.223 | 0.215 |
| Number of older patients living alone | 256 | 218 | 285 | 0.113 | 0.356 | 0.612 |
| Number of older patients requiring help in daily activity | 1895 | 2001 | 2114 | 0.102 | 0.166 | 0.209 |
| Total number of prescribed medicines | 14823 | 15923 | 14258 | 0.110 | 0.112 | 0.095 |
| Total number of non-prescribed medicines | 6623 | 6412 | 6212 | 0.211 | 0.358 | 0.451 |
| Total number of dispensed medicines | 21446 | 22335 | 20470 | 0.169 | 0.118 | 0.091 |
| Total number of older patients with DRPs | 214 | 689 | 318 | **0.023** | 0.076 | **0.034** |
| Total DRPs | 428 | 2411 | 782 | **0.012** | **0.035** | **0.035** |
| Incidence of DRPs (total DRPs/total number of dispensed medicines ×100) | 2.0% | 10.8% | 3.8% | **0.011** | 0.069 | **0.014** |
| DRPs per patients | 0.1 | 0.7 | 0.2 | **0.032** | 0.170 | **0.041** |
| **DRP classification** | | | | | | |
| Avoidable medicine | 28 (6.5%) | 355 (14.7%) | 79 (10.1%) | **0.041** | 0.072 | 0.081 |
| Untreated disease | 31 (7.2%) | 221 (9.2%) | 68 (8.7%) | | | |
| Need for additional treatment | 45 (10.5%) | 184 (7.6%) | 57 (7.3%) | | | |
| Low dose | 40 (9.3%) | 172 (7.1%) | 79 (10.1%) | | | |
| Drug-Drug interaction | 39 (9.1%) | 186 (7.7%) | 65 (8.3%) | | | |
| High dose | 39 (9.1%) | 184 (7.6%) | 58 (7.4%) | | | |
| Contraindication | 41 (9.6%) | 191 (7.9%) | 85 (10.9%) | | | |
| A potential risk for adverse drug reaction | 29 (6.8%) | 169 (7.0%) | 64 (8.2%) | | | |
| Allergy | 19 (4.4%) | 174 (7.2%) | 45 (5.8%) | | | |
| Poor knowledge | 43 (10.0%) | 265 (11.0%) | 69 (8.8%) | | | |
| Poor adherence | 33 (7.7%) | 144 (6.0%) | 71 (9.1%) | | | |
| Necessity for monitoring | 41 (9.6%) | 166 (6.9%) | 42 (5.4%) | | | |
| **DRP severity** | | | | | | |
| Mild | 284 (66.4%) | 1356 (56.2%) | 446 (57.0%) | 0.103 | 0.098 | 0.233 |
| Significant | 107 (25.0%) | 813 (33.7%) | 269 (34.4%) | | | |
| Serious | 32 (7.5%) | 196 (8.1%) | 55 (7.0%) | | | |
| Life-threatening | 5 (1.2%) | 46 (1.9%) | 12 (1.5%) | | | |
| **Evidence quality** | | | | | | |
| Poor | 127 (29.7%) | 426 (17.7%) | 149 (19.1%) | 0.122 | 0.108 | 0.331 |
| Moderate | 196 (45.8%) | 1120 (46.5%) | 368 (47.0%) | | | |
| High | 105 (24.5%) | 865 (35.5%) | 265 (33.9%) | | | |

DRP: drug-related problem.

P value[a]; indicates the significance of differences across the control and intervention group A.

P value[b]; indicates the significance of differences across the control and intervention group B.

P value[c]; indicates the significance of differences across the intervention group A and intervention group B. Data are presented as numbers with proportions unless stated otherwise. A p-value of less than 0.05 was considered a significant result

Additionally, the proportions of acceptance of pharmacist interventions at the patient level were similar across the study groups (p = 0.118). However, the proportion of pharmacist interventions that were accepted by physicians in Active Group A (58.6%) was significantly higher than that of the control group (50.9%, p = 0.041), and Active Group B (38.8%, p = 0.012).

The mean cost reduction per patient in Active Group A (31.3±11.8 $) was significantly higher than that in Active Group B (20.8±8.6 $) and the control group (19.6±9.5 $), as shown in Fig 2. Our sub-analysis revealed that the means of cost reduction per patient were

**Table 3. Patterns, classification, and clinical significance of pharmacist interventions.**

| Pharmacist interventions | Control group Total (N = 394) | Accepted (N = 214) | Not accepted (N = 180) | Intervention group A Total (N = 1629) | Accepted (N = 943) | Not accepted (N = 986) | Intervention group B Total (N = 536) | Accepted (N = 267) | Not accepted (N = 222) | p-value |
|---|---|---|---|---|---|---|---|---|---|---|
| **Pharmacist intervention at the physician level** | 171 (43.4%) | 87 (50.9%) | 84 (49.1%) | 769 (47.2%) | 451 (58.6%) | 318 (41.4%) | 201 (37.5%) | 78 (38.8%) | 123 (61.2%) | **0.012**[a] 0.086[b] **0.041**[c] **0.019**[d] **0.012**[e] |
| Recommend cessation of a medication | 33 (19.3%) | 19 (57.6%) | 14 (42.4%) | 286 (37.2%) | 178 (62.2%) | 108 (37.8%) | 65 (32.3%) | 28 (43.1%) | 37 (56.9%) | |
| Recommend decreasing a dose | 51 (29.8%) | 35 (68.6%) | 16 (31.4%) | 146 (19.0%) | 81 (55.5%) | 65 (45.5%) | 38 (18.9%) | 14 (36.8%) | 24 (63.2%) | |
| Recommend increasing a dose | 25 (14.6%) | 9 (36.0%) | 16 (64.0%) | 74 (9.6%) | 36 (48.6%) | 38 (51.4%) | 28 (13.9%) | 12 (42.9%) | 16 (57.1%) | |
| Recommend changing a medication | 35 (20.5%) | 11 (31.4%) | 24 (68.6%) | 150 (19.5%) | 96 (64.0%) | 54 (36.0%) | 52 (25.9%) | 16 (30.8%) | 36 (69.2%) | |
| Recommend specific laboratory tests | 20 (11.7%) | 10 (50.0%) | 10 (50.0%) | 65 (8.5%) | 33 (50.8%) | 32 (49.2%) | 11 (5.5%) | 5 (45.5%) | 6 (54.5%) | |
| Recommend other medical procedures | 7 (4.1%) | 3 (42.9%) | 4 (57.1%) | 48 (6.2%) | 27 (56.3%) | 21 (43.7%) | 7 (3.5%) | 3 (42.9%) | 4 (57.1%) | |
| **Pharmacist intervention at the patient level** | 223 (56.6%) | 127 (57.0%) | 96 (43.0%) | 860 (52.8%) | 492 (57.2%) | 368 (42.8%) | 335 (62.5%) | 189 (56.4%) | 146 (43.6%) | 0.118[a] 0.094[b] 0.631[c] 0.322[d] 0.462[e] |
| Referral to a prescriber | 31 (13.9%) | 18 (58.1%) | 13 (41.9%) | 114 (13.3%) | 52 (45.6%) | 62 (54.4%) | 37 (11.0%) | 18 (48.6%) | 19 (51.4%) | |
| Referral to a hospital | 18 (8.1%) | 7 (38.9%) | 11 (61.1%) | 74 (8.6%) | 17 (23.0%) | 57 (77.0%) | 25 (7.5%) | 8 (32.0%) | 17 (68.0%) | |
| Patient education | 81 (28.3%) | 59 (72.8%) | 22 (27.2%) | 251 (29.2%) | 196 (78.1%) | 55 (21.9%) | 119 (35.5%) | 92 (77.3%) | 27 (22.7%) | |
| Recommend a monitoring plan | 26 (11.7%) | 13 (50.0%) | 13 (50.0%) | 75 (8.7%) | 33 (44.0%) | 42 (56.0%) | 41 (12.2%) | 14 (34.1%) | 27 (65.9%) | |
| Recommend seeking family help | 20 (9.0%) | 9 (45.0%) | 11 (55.0%) | 41 (4.8%) | 16 (39.0%) | 25 (61.0%) | 23 (6.9%) | 9 (39.1%) | 14 (60.9%) | |
| Patient persuasion to use telepharmacy | 18 (8.1%) | 7 (38.9%) | 11 (61.1%) | 65 (7.6%) | 38 (58.5%) | 27 (41.5%) | 30 (9.0%) | 13 (43.3%) | 17 (56.7%) | |
| Patient persuasion to do exercise | 10 (4.5%) | 6 (60.0%) | 4 (4.0%) | 50 (5.8%) | 16 (32.0%) | 24 (48.0%) | 21 (6.3%) | 11 (52.4%) | 10 (47.6%) | |
| Patient persuasion to eat healthy food | 10 (4.5%) | 5 (50.0%) | 5 (5.0%) | 51 (5.9%) | 26 (51.0%) | 25 (49.0%) | 19 (5.7%) | 8 (42.1%) | 11 (57.9%) | |
| Ensure the patient knows how to ask for emergency care | 9 (4.0%) | 7 (77.8%) | 2 (22.2%) | 104 (12.1%) | 98 (94.2% | 6 (5.8%) | 20 (6.0%) | 16 (80.0%) | 4 (20.0%) | |

P-value[a] was calculated to measure differences in the proportion of acceptance of pharmacist interventions at physician and patient levels across the study groups.

P-value[b] was calculated to measure differences in the categories of pharmacist interventions across the study groups.

P-value[c] was calculated to measure differences in the proportion of acceptance of pharmacist interventions at physician and patient levels between the control group and the intervention group A.

P-value[d] was calculated to measure differences in the proportion of acceptance of pharmacist interventions at physician and patient levels between the control group and the intervention group B.

P-value[e] was calculated to measure differences in the proportion of acceptance of pharmacist interventions at physician and patient levels between the intervention group A and the intervention group B.

significantly higher in patients with major polypharmacy among all study groups (p<0.05), as depicted in Fig 3. Additionally, the mean time needed to resolve a drug-related problem (DRP) in Active Group A (7.3±3.5 minutes) was significantly lower than that in Active Group B (9.8±4.2 minutes) and the control group (9.8±5.7 minutes), as illustrated in Fig 4.

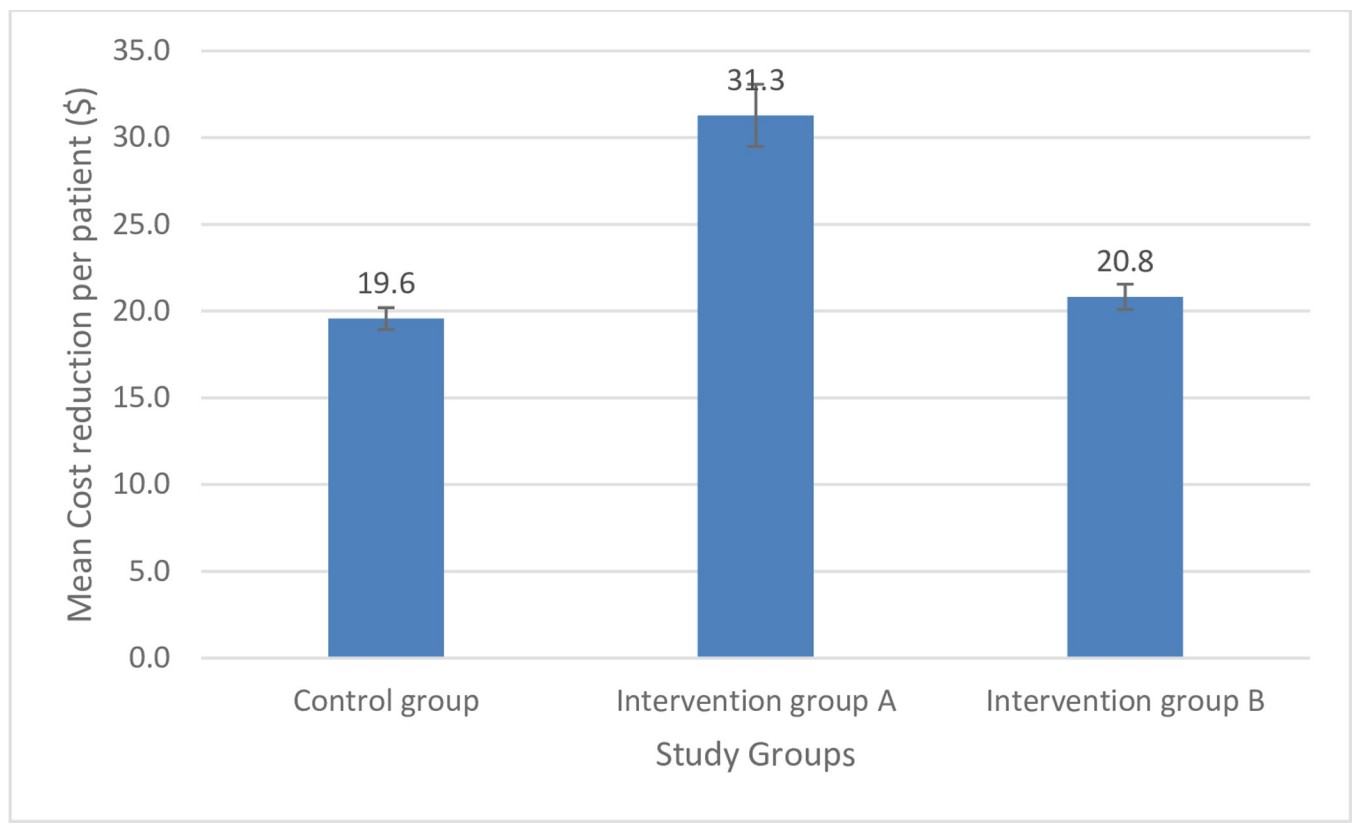

**Fig 2. The mean cost reduction per patient across the study groups.**

## Discussion

Elderly individuals are more susceptible to drug-related problems (DRPs) and medication errors, a vulnerability that has been exacerbated by the COVID-19 pandemic and its impacts on healthcare accessibility and the shortage of medical personnel [26]. Pharmacists, however, are well-equipped to optimize medication use in elderly, but they must continuously educate themselves and adapt to new challenges [27]. Traditional educational interventions, such as workshops and lectures, are often time-consuming, costly, and suffer from low engagement from learners [28].

In this study, we aimed to evaluate the effectiveness of two unique educational interventions on pharmacists' ability to optimize medication therapy plans in elderly. Specifically, we delivered clinical cases through WhatsApp and research articles via email to community pharmacists, and measured their interventions on elderly. As far as we know, this is the first study to investigate this type of educational intervention on pharmacists' ability to identify DRPs in elderly.

Our research findings showed that the delivery of clinical cases via WhatsApp significantly improved pharmacists' ability to identify drug-related problems (DRPs) in elderly. In contrast, sending lengthy research articles through email did not result in a noteworthy increase in the number of DRPs identified by pharmacists [29, 30]. Furthermore, pharmacists who received clinical case scenarios through WhatsApp took less time to resolve a DRP and achieved a greater reduction in the cost compared to other groups. This can be attributed to the informative and concise nature of the real-life clinical scenarios delivered through WhatsApp. Moreover, the use of this platform provided pharmacists with a unique educational experience,

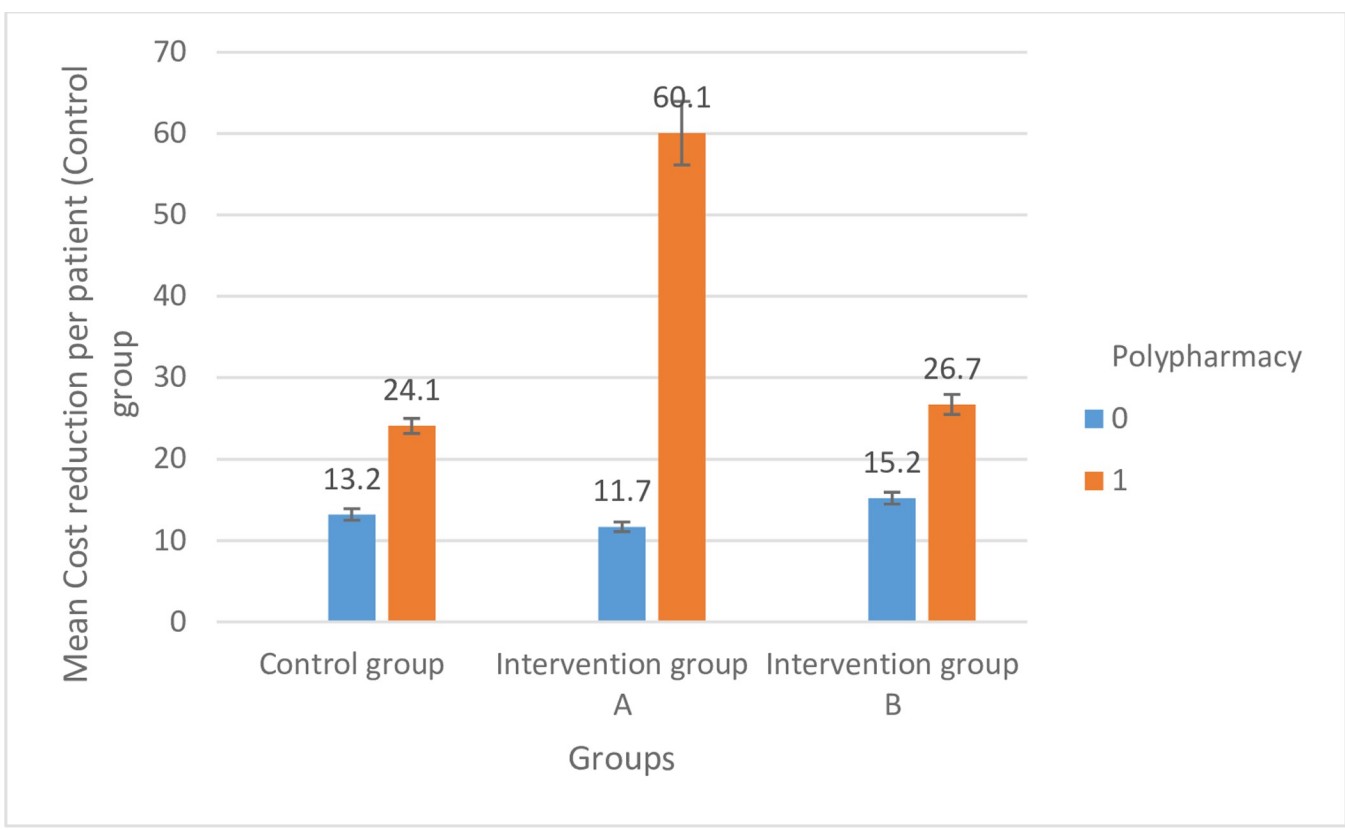

**Fig 3. The association between polypharmacy and cost reduction across the study groups (0: No major polypharmacy; 1: Major polypharmacy).**

removing communication barriers with educators and potentially enhancing their engagement in educational programs. Conversely, pharmacists may face difficulties in reading lengthy articles to extract relevant information about elderly care and safety due to their heavy workloads.

In this study, pharmacists in Active Group A identified significantly different types of DRPs compared to those detected by pharmacists in the control group. The pharmacists in Active Group A were particularly successful in identifying more avoidable medicine-based DRPs. As a result, the most common intervention made by pharmacists in Active Group A was to recommend the cessation of medication. This highlights that the educational intervention provided to Active Group A could improve pharmacists' deprescribing skills. Deprescribing, or discontinuing unnecessary or harmful medication, has been shown to reduce the risk of falls [31] and functional decline [32] in elderly.

Despite being presented with clinical scenarios requiring deprescribing of potentially inappropriate medications, Active Group A pharmacists were not trained in using standard deprescribing protocols, such as STOPP and Beers criteria [21, 33]. This highlights the need for further research into the appropriateness of deprescribing interventions made by pharmacists [30, 34]. While some studies have evaluated the impact of education on pharmacists' ability to detect and address DRPs [35, 36], there is still a limited amount of research in this area. Most previous studies have only focused on the frequency and types of pharmacist interventions, without attempting to improve the effectiveness of these interventions. For instance, Mekdad and Alsayed [37] found a high prevalence of DRPs in elderly and demonstrated that pharmacist interventions can reduce the risk of negative clinical outcomes. Ali et al [38] conducted a systematic review of 3826 studies and found that pharmacist-led interventions have the potential to decrease the occurrence of DRPs.

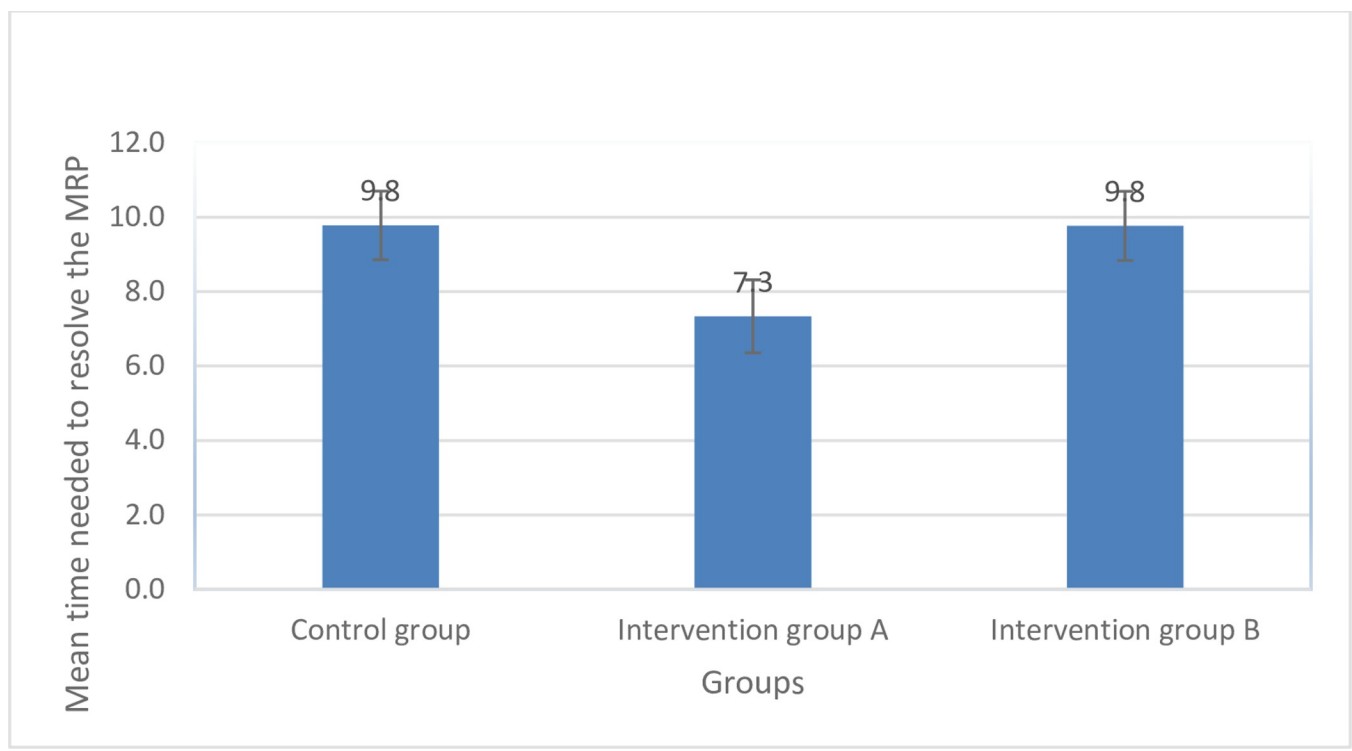

**Fig 4. Differences in the mean time needed to resolve the MRP across the study groups.**

Despite similar levels of severity of DRPs and evidence quality among the study groups, the rate of physician acceptance of pharmacist interventions was higher in Active Group A. The higher rate of physician acceptance of pharmacist interventions in Active Group A could be attributed to nature of clinical scenarios provided to pharmacists in this group. This education may have improved pharmacists' ability to communicate effectively with physicians and present recommendations in a manner that is more likely to be accepted. Further research should be conducted to explore the impact of educational interventions on physician acceptance of pharmacist interventions and the potential for improvement in the quality of patient care. Previous studies demonstrated that a successful pharmacist intervention often depends on physician acceptance and collaboration [39, 40]. Physician acceptance is an important factor in determining the success of pharmacist interventions, as it directly influences the implementation and adherence to recommendations.

## Limitations

This study had a few limitations that need to be considered when interpreting the results. The main limitation is the use of self-reporting as the method of assessment. Self-reporting can be subject to intentional or unintentional bias [41], which may have affected the accuracy of the findings. However, the use of WhatsApp as the platform for self-reporting helped reduce the risk of bias by allowing for a blind communication between investigators and participants. Another limitation is the focus on process-based DRPs and pharmacist interventions without considering outcome-based parameters. Outcome-based parameters are crucial in determining the efficacy of a particular approach, as they assess the actual impact on patient outcomes [42]. The study also did not examine the impact of educational interventions on physician acceptance of pharmacist recommendations. Further research is critical to improve our understanding of these limitations and how they impact the results.

## Conclusion

The implementation of WhatsApp as a tool for delivering clinical scenarios has demonstrated effectiveness in enhancing pharmacists' ability in identifying DRPs in elderly patients. This approach has also demonstrated a reduction in the time required to address DRPs and an increase in cost savings per patient. However, additional studies with a focus on outcome-based parameters are required to evaluate the effect of this intervention on the clinical value of pharmacist recommendations. Furthermore, the influence of this approach needs to be studied in the context of pharmacist-physician relationship.

## Supporting information

**S1 File. CONSORT 2010 checklist of information to include when reporting a randomised trial\*.**
(DOC)

## Acknowledgments

We would like to acknowledge the following universities: New Giza University-Egypt, University of Sharjah-UAE; Al Ain University-Abu Dhabi-UAE, University of Petra-Jordan, Gulf Medical University-Ajman-UAE, National University-Sudan, University of Almaarefa-(Diriyah)-Riyadh-Saudi Arabia, Taif University-Saudia Arabia, Imam Abdulrahman Bin Faisal University-Saudi Arabia, and Dar Al Uloom University- Riyadh-Saudi Arabia. Our great thanks go to Rawan Khalosh for her esteemed efforts.

## Author Contributions

**Conceptualization:** Osama Mohamed Ibrahim, Nadia Al Mazrouei, A. A. Elnour, Rana Ibrahim, Derar H. Abdel-Qader, Rowida Mohamed El Amin Ibrahim Hamid, Vineetha Menon, Ali Awadallah Saeed, Sami Fatehi Abdalla.

**Resources:** Fahad T. Alsulami, Yousef Saeed Alqarni, Abuelnor Mohammed.

**Writing – original draft:** Osama Mohamed Ibrahim, Nadia Al Mazrouei, A. A. Elnour, Rana Ibrahim, Derar H. Abdel-Qader, Rowida Mohamed El Amin Ibrahim Hamid, Vineetha Menon, Ali Awadallah Saeed, Sami Fatehi Abdalla, Fahad T. Alsulami, Yousef Saeed Alqarni, Abuelnor Mohammed.

**Writing – review & editing:** Osama Mohamed Ibrahim, Nadia Al Mazrouei, A. A. Elnour, Rana Ibrahim, Derar H. Abdel-Qader, Rowida Mohamed El Amin Ibrahim Hamid, Vineetha Menon, Ali Awadallah Saeed, Sami Fatehi Abdalla, Fahad T. Alsulami, Yousef Saeed Alqarni, Abuelnor Mohammed.

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
