## [Decision Letter · Decision Letter 0]

7 Jun 2024

PONE-D-24-03132Randomized Controlled Trial on Optimizing Community Pharmacist’s Care for the Elderly: The Influence of WhatsApp-Email delivered Clinical Case ScenariosPLOS ONE

Dear Dr. Saeed,

Thank you for submitting your manuscript to PLOS ONE. After careful consideration, we feel that it has merit but does not fully meet PLOS ONE’s publication criteria as it currently stands. Therefore, we invite you to submit a revised version of the manuscript that addresses the points raised during the review process.

We look forward to receiving your revised manuscript.

Kind regards,

Naeem Mubarak, PhD

Academic Editor

PLOS ONE

5. Please include your tables as part of your main manuscript and remove the individual files. Please note that supplementary tables (should remain/ be uploaded) as separate "supporting information" files.

Additional Editor Comments:

The manuscript has been critically reviewed and requires major revisions due to concerns raised by the reviewer on the recruitment of intervention and control groups, sample size calculations and overall analysis of the findings. Addressing these concerns can improve the quality of evidence intended to be established through this Randomized Controlled Trial.

Reviewers' comments:

Reviewer's Responses to Questions

**Comments to the Author**

1. Is the manuscript technically sound, and do the data support the conclusions?

Reviewer #1: No

2. Has the statistical analysis been performed appropriately and rigorously? 

Reviewer #1: No

3. Have the authors made all data underlying the findings in their manuscript fully available?

Reviewer #1: Yes

4. Is the manuscript presented in an intelligible fashion and written in standard English?

Reviewer #1: No

5. Review Comments to the Author

Reviewer #1: Thank you for providing me a valuable opportunity to review this randomized controlled trial. The study focused on assessing the impact of educational interventions on pharmacists’ ability to identify DRPs in elderly population. These educational interventions used WhatsApp and email sent clinical case scenarios to the community pharmacists. The study has certainly intended to contribute meaningfully in educating community pharmacists, however, has certain deficiencies and concerns which need to be addressed by the authors to further improve the quality, clarity and essence of the manuscript. Following are the recommendations for the authors to consider:

Background/Rationale of the study:

• The authors mention “DRPs are prevalent among older populations, with rates ranging from 34-56%”. Are these statistics reported from worldwide or only from UAE? Please clarify.

• The rationale of the study intends to observe the impact of these educational interventions on the practices of the community pharmacists while dealing with elderly patients, however, the manuscript does not highlight any further of how this impact was determined?

• Was there any practical observation of those community pharmacists in dealing with the elderly population after the dissemination of these educational interventions? Or was the identification of drug related problems assessed from the 24 clinical scenarios. Please justify.

• “The second intervention is the use of lengthy articles related to older patient care sent via email” How would the authors define “lengthy articles”

• “The study will examine the impact of these interventions on pharmacists’ ability to identify drug-related problems, perform medication reviews, and perform pharmaceutical intervention”. The article sheds little detail on how this objective was achieved.

• The clinical scenarios are understandable in providing pharmacists with the necessary education and assessing their ability to identify DRPs, however, how this was accomplished through research articles is not documented enough. Please justify.

Methodology:

• What sampling method was employed to collect data from 127 pharmacies?

• How the data for the population size of total community pharmacies in UAE were gathered? Based on this, how the minimum sample size for both community pharmacies and community pharmacists was determined?

• The authors used a different mediums (WhatsApp and email) for clinical case scenarios and research articles. Would this not affect the response received via email? How have the authors concluded that the low response for email sent articles was due to their length and not due to using email as a medium? Were these confounding factors taken into consideration?

• “The cases, with a maximum word count of 150, were based on the panel members' experiences and previous literature”. No reference has been cited here to indicate the literature used to prepare 24 clinical case scenarios

• How the reliability and validity of these clinical cases were ensured?

• “The primary outcome measure was the improvement in the ability to identify DRPs”. Please consider mentioning the criteria for this improvement. Also was this improvement noted during the reporting of drug related problems from the clinical case scenarios or elderly patients coming to the community pharmacies after receiving these interventions? Please justify.

• For control, Group C, the authors mention no detail of how the outcome was measured from them. For instance, to compare with the study groups, it is assumed that they did not receive any clinical case scenarios and research articles. Therefore, how the competency of community pharmacists in this group was determined still entails comprehensive justification and explanation.

• It is difficult to comprehend how the authors have determined the cost reduction per patient. Whether the cost reduction was determined by identifying the drug related problem in clinical case scenarios and then suggesting deprescribing of some medications leading to reduced medications and overall costs. If that is the case, how can we associate it with the impact of educational interventions afterwards? Please justify.

• “The severity of DRPs was evaluated by a multidisciplinary expert panel, which met twice a month during the data reporting period and reviewed and rated each DRP reported by the participants” What was the rating criteria? Please clarify

Results:

• A total of 468 community pharmacists working in 131 pharmacies agreed to participate in the study and were assigned to three groups; the control group, Active Group A, and Active Group B. In what ratio, these pharmacists were allocated in these three groups? Please clarify. How selection and allocation biases were prevented in this case?

• Although the authors have provided the statistics of the findings however, still they are difficult to comprehend whether they were measured through the clinical case scenarios or on elderly patients coming to community pharmacies. As the objective of measuring the impact of educational interventions appears vague if the same clinical scenarios and research articles are used for intervention and outcome measurement. This is the major reservation found throughout the manuscript which must be justified in a comprehensive manner.

Discussion:

• The discussion section needs further improvement to present a detailed analysis of the findings. The findings should be compared with the results from other countries.

• To update references on physicians’ acceptance of pharmacists’ recommendations and direct towards interprofessional collaboration, the following studies may be cited (This is optional and should only be taken as a suggestion for the improvement of the manuscript)

1. DOI: 10.3389/fpubh.2024.1323102

2. DOI: 10.2147/RMHP.S296113

• The strengths, limitations and future prospects of the study must be mentioned comprehensively. More elaboration is required on how the authors have minimized different kinds of bias associated with RCTs such as selection bias, allocation bias, attrition bias, confounding bias etc.

References:

• Several of the references need to be updated. Please check throughout.

While the manuscript offers understanding on the crucial topic of pharmacist’s ability and competency in dealing with elderly patients, it lacks in methodological rigor and clarity in presentation of the findings.

6. PLOS authors have the option to publish the peer review history of their article (what does this mean?). If published, this will include your full peer review and any attached files.

Reviewer #1: No

---

## [Author Response · Author response to Decision Letter 0]

8 Jul 2024

Dear All

A rebuttal letter that responds to each point raised by the academic editor and reviewer uploaded.

---

## [Decision Letter · Decision Letter 1]

24 Jul 2024

Randomized Controlled Trial Parallel-Group on Optimizing Community Pharmacist’s Care for the Elderly: The Influence of WhatsApp-Email delivered Clinical Case Scenarios

PONE-D-24-03132R1

Dear Dr. Ali Awadallah Saeed,

We’re pleased to inform you that your manuscript has been judged scientifically suitable for publication and will be formally accepted for publication once it meets all outstanding technical requirements.

Kind regards,

Naeem Mubarak, PhD

Academic Editor

PLOS ONE

Additional Editor Comments (optional):

The manuscript has addressed all the concerns raised by the reviewer, hence no further recommendations are required. The manuscript may be accepted for publication. Best of luck with your publication!

Reviewers' comments:

Reviewer's Responses to Questions

**Comments to the Author**

1. If the authors have adequately addressed your comments raised in a previous round of review and you feel that this manuscript is now acceptable for publication, you may indicate that here to bypass the “Comments to the Author” section, enter your conflict of interest statement in the “Confidential to Editor” section, and submit your "Accept" recommendation.

Reviewer #1: All comments have been addressed

2. Is the manuscript technically sound, and do the data support the conclusions?

Reviewer #1: Yes

3. Has the statistical analysis been performed appropriately and rigorously? 

Reviewer #1: Yes

4. Have the authors made all data underlying the findings in their manuscript fully available?

Reviewer #1: Yes

5. Is the manuscript presented in an intelligible fashion and written in standard English?

Reviewer #1: Yes

6. Review Comments to the Author

Reviewer #1: All the comments have been addressed by the authors. No further improvements or suggestions required.

7. PLOS authors have the option to publish the peer review history of their article (what does this mean?). If published, this will include your full peer review and any attached files.

Reviewer #1: No

---

## [Editor Report · Acceptance letter]

13 Aug 2024

PONE-D-24-03132R1 

PLOS ONE

Dear Dr. Saeed, 

I'm pleased to inform you that your manuscript has been deemed suitable for publication in PLOS ONE. Congratulations! Your manuscript is now being handed over to our production team.

Kind regards, 

on behalf of

Dr Naeem Mubarak 

Academic Editor

PLOS ONE